# Adaptation and validation of simple tools to screen and monitor for oral PrEP adherence

Seth Zissette[1,2], Elizabeth E. Tolley[1]*, Andres Martinez[1], Homaira Hanif[3], Katherine Gill[4], Nelly Mugo[5], Laura Myers[4], Ednar Casmir[5], Menna Duyver[4], Kenneth Ngure[6], Gustavo F. Doncel[3]

**1** FHI 360, Behavioral, Epidemiological, Clinical Sciences, Durham, North Carolina, United States of America, **2** University of Notre Dame, Wilson Sheehan Lab for Economic Opportunities, Notre Dame, Indiana, United States of America, **3** CONRAD, Eastern Virginia Medical School, Norfolk, Virginia, United States of America, **4** Desmond Tutu HIV Foundation, University of Cape Town, Cape Town, South Africa, **5** Center for Clinical Research (CCR), Kenya Medical Research Institute (KEMRI), Nairobi, Kenya, **6** Department of Community Health, Jomo Kenyatta University of Agriculture and Technology, Nairobi, Kenya

* btolley@fhi360.org

## Abstract

### Introduction

Oral, vaginal and other pre-exposure prophylaxis (PrEP) products for HIV prevention are in various stages of development. Low adherence poses a serious challenge to successful evaluation in trials. In a previous study, we developed tools to screen for general adherence and specifically monitor intravaginal ring adherence within the context of HIV prevention clinical trials. This study aimed to further validate the screening tool and to adapt and provide initial psychometric validation for an oral pill monitoring tool.

### Materials and methods

We administered a cross-sectional survey between June and October 2018 at a trial site located near Cape Town, South Africa, and another in Thika, Kenya, with 193 women who had experience using daily oral pills. We fit confirmatory factor analysis models on the screening tool items to assess our previously-hypothesized subscale structure. We conducted an exploratory factor analysis of oral PrEP monitoring items to determine the underlying subscale structure. We then assessed the construct validity of each tool by comparing subscales against each other within the current sample and against our original sample, from a study conducted in four sites in South Africa, including Cape Town.

### Results

The screening tool structure showed moderate evidence of construct validity. As a whole, the tool performed in a similar way to the original sample. The monitoring tool items, which were revised to assess perceptions about and experiences using daily oral PrEP, factored into five subscales that showed moderate to good reliability. Four of the five subscales had a similar structure overall to the vaginal ring monitoring tool from which they were adapted.

**Data Availability Statement:** The data can be found on the Harvard Dataverse Repository at https://doi.org/10.7910/DVN/NI4ZQA.

**Funding:** This study was funded by the U.S. President's Emergency Plan for AIDS Relief

(PEPFAR) and the U.S. Agency for International Development (USAID) through a cooperative agreement (AID-OAA-A-14-00011) with CONRAD and Eastern Virginia Medical School. The funders had no role in study design, data collection and analysis, decision to publish, or preparation of the manuscript.

**Competing interests:** The authors have declared that no competing interests exist.

## Conclusions

Accurate measurement of HIV-prevention product adherence is of critical importance to the assessment of product efficacy and safety in clinical trials, and the support of safe and effective product use in non-trial settings. In this study, we provide further validation for these measures, demonstrating the screening tool's utility in additional populations and adapting the monitoring tool's utility for different HIV-prevention products.

## Introduction

New biomedical methods for HIV prevention have changed the landscape of the epidemic over the past decade. Daily oral regimens of pre-exposure prophylaxis (PrEP) have proven effective in reducing transmission [1–3] and been introduced in multiple settings [4, 5]. Additional HIV-only and multipurpose prevention products, such as vaginal rings, long-acting injectables, implants, and HIV vaccines are at various stages in the development pipeline as well; as of October 2019, the AIDS Vaccine Advocacy Coalition identified over 50 ongoing or planned HIV prevention product trials [6, 7]. Over one third of these are focused primarily or significantly on use by women, who often face disproportionate burdens of the disease and unique challenges to utilizing prevention products [6, 8].

As PrEP and other HIV prevention products become more widely available, product adherence poses a serious challenge to their successful use, particularly among young African women [6, 8, 9]. In trials of PrEP effectiveness, sub-analyses indicated that poor adherence to the prescribed dosing regimen reduced effectiveness [10, 11]. The importance of adherence to product use instructions was underscored by results from the VOICE and FEM-PrEP trials, where poor adherence to vaginal (VOICE) and oral (VOICE and FEM-PrEP) dosing regimens undermined the trials' ability to determine effectiveness [12, 13]. Validated tools that enable trial implementers to proactively screen for and prospectively monitor adherence-related determinants are vitally important for current and future studies.

Conventional self-reported adherence instruments within HIV prevention trials often rely on single-item questions. While inexpensive and easy to administer, they are prone to self-presentation and/or recall biases or require participants to make complex mental calculations [14, 15]. Consequently, HIV prevention trials generally prefer to use objective markers (e.g., detection of drug presence or level in blood or pill count measures) to assess product adherence [16, 17]. However, these measures also have limitations [18]. For example, biomarker measures are often expensive, may be prone to a "white coat effect" and generally measure adherence to active, but not placebo products [19]. Although studies have found some evidence of dose-response between adherence and residual drug levels in returned vaginal rings, individual variability challenges this as an approach to monitoring adherence [20]. Drug concentrations along strands of hair offer a more promising biomarker of cumulative adherence [21, 22]. Pill counts may over or under-estimate adherence due to pill-dumping or use of pill cases [23]. Scales are composed of multiple items in the form of questions or statements that, when combined, measure a more complex construct that may not be directly observable [24]. When psychometrically validated, such measures may better assess the multiple factors contributing to adherence/non-adherence [25–28].

In response to the measurement challenges cited above, we previously conducted a study in four South African sites [9] to develop and provide preliminary validation for two different tools to screen for and monitor product-related adherence within the context of clinical trial

participation. The screening tool comprised six subscales measuring potential participants' attitudes about, and social and structural challenges towards trial participation. Items were intended to be administered prior to product use and were product-agnostic. The monitoring tool comprised four sets of items intended to measure specific attitudes related to using product over trial participation. In the previous study, three versions of the monitoring items were developed relating to use of a vaginal ring, vaginal gel and oral pills. Although the vaginal ring items were validated, the survey sample was insufficiently powered to validate the vaginal gel or oral pill monitoring items. Past research has shown the degree to which product type can influence product acceptability and adherence [29–31], necessitating the need for product-specific adaptations for adherence monitoring. In order to facilitate the use of the screening and monitoring tools in upcoming trials of novel antiretroviral (ARV) oral pills, we sought to refine and validate both the screening tool and the oral pill monitoring tool items.

The overall goal of this project was to adapt and provide further psychometric validation for the screening tool and initial psychometric validation for the oral pill monitoring tool. Starting with the tools developed in the aforementioned study (from here forward referred to as the "original" study) [9], we used scale development processes to adapt the screening and monitoring tool items for local context in two sites in South Africa and Kenya, including rounds of cognitive interviewing to improve item framing, comprehensibility and salience, followed by administration of a survey to facilitate both confirmatory and exploratory factor analyses and psychometric evaluation of resulting constructs.

## Materials and methods

### Data collection

We administered a cross-sectional survey between June and October 2018 in two trial sites located near Cape Town, South Africa, and Thika, Kenya, both with prior experience implementing PrEP clinical trials. Prior to implementing the survey, we conducted cognitive interviews with 36 women (18 each from Cape Town, South Africa and Thika, Kenya) to assess whether the screening and monitoring items from our original study were salient, relevant and easy to respond to. The study protocol was approved by FHI360's and local IRBs in both countries. The survey was administered face-to-face by data collectors who electronically captured responses on tablets with a total of 193 women. Participants were eligible for recruitment for either the cognitive interviews or the survey if they were female, between the ages of 18–30, had experience using daily oral pills (any indication), and provided written consent electronically prior to the survey. Because potential trial participants for whom our scales are intended might comprise those with and without previous experience in clinical research, we recruited both former trial participants (FTP) from previous or ongoing trials of oral PrEP and trial-naïve participants (TNP). Study recruitment was conducted by trained clinical trial outreach staff who consulted logs of former participants with consent for recontact and worked through local health clinics and other groups to recruit trial-naïve participants. Participants could elect to hear and respond to any part of or all of the survey in either English or their local language (isiXhosa or Kiswahili).

The survey contained three primary sections. Following basic demographic and past trial experience questions to determine eligibility, the first section included 54 items in random order making up the six subscales of the screening tool. The same items were administered uniformly to all women. The second section comprised of 45 monitoring items in random order adapted for oral PrEP. There were two versions of this section; one focused on oral PrEP and administered only to women who had or were participating in oral PrEP trials, the other version adapted oral PrEP-specific items to other types of oral pill use to better fit the

experiences of trial-naïve women. All items in the screening tool and monitoring tool sections were asked on a 6-point Likert scale with responses ranging from "1 = Disagree a lot", "2 = Disagree somewhat", "3 = Disagree a little" to "4 = Agree a little", "5Agree = Agree somewhat", and "6 = Agree a lot". The third section included additional demographic and psychosocial variables, such as age, education level, income status, and relationship status and dynamics.

## Data analysis

**Screening tool.**   Our original tool, developed through research in four sites in South Africa [9], comprised a total of 59 items organized into six domains, which consisted of factors that measured reasons for clinical trial participation (Commitment to Research, Personal & Health Benefits), beliefs about the trial and product (Distrust of Research) and agency (Trial Incompatibility, Partner Disclosure, Visit Adherence). Because the tool was intended to be administered prior to a trial participant's randomization to product, the items are product agnostic. Therefore, we aimed to further validate the tool by determining how similarly the items performed in our new study samples relative to our original study.

Following cognitive interviews with 36 women to contextually adapt the items, we retained 54 items relevant in both settings for use in the survey. Using only these items from the survey data, we started by fitting a confirmatory factor analysis (CFA) model reflecting the hypothesized 6-factor structure to the full new sample of participants. We then repeated the analysis using only the sample of former-trial participants. Rather than treat items as continuous as is common in standard CFAs, we treated the response scale of all the items as ordinal to reflect its Likert nature.

**Monitoring tool.**   Our original monitoring tool comprised of 48 vaginal ring-specific items organized into 4 domains, relating to vaginal ring doubts, benefits, side effects and removal. We assumed that the new and/or revised items relating to oral pill adherence might not fit the same thematic structure as the earlier vaginal ring items.

For this analysis, we used only the 45 monitoring items included in the survey. We examined the response distribution, means, standard deviations, skew, and kurtosis of all items, with the aim of eliminating items that exhibited high skew or little variation as evidenced by flooring/ceiling effects, low standard deviation, or high skew/kurtosis. We then conducted exploratory factor analyses (EFA) of monitoring items. We ran an EFA with oblique (Promax) rotation and an unrestricted number of factors to determine the possible number of factors. We then examined the scree plot and table of eigenvalues to identify the number of factors with eigenvalues greater than 1, as well as observing the least diminishing difference between next eigenvalue, to assess how many factors should be extracted.

After determining a possible acceptable range for number of factors, we generated each possible solution iteratively. This involved the following steps:

- Conducting EFAs with oblique (Promax) rotation and dropping items with loadings less than 0.4 for each possible solution with the specified number of factors.

- Dropping any items that did not load to any factor in any solution and re-running EFAs until all included items loaded onto at least one factor in at least one solution.

- Assessing reliability using Cronbach's alpha with a preferred threshold of 0.7 and content validity of each factor in each solution.

- Comparing factor solutions with previous vaginal ring monitoring subscales to determine the best possible solution.

**Construct validity.** We assessed tool validity using a multifaceted approach. We first compared the final proposed screening tool and monitoring tool against their original versions, focusing on reliability and number of items. We then compared Pearson Product Moment correlations between the screening tool and monitoring tool within and between the original and current samples to determine whether the scales were performing in a predictable manner. We considered differences of 0.20 and higher to indicate relationships between subscales that differed from the original sample. Finally, we examined descriptive statistics for subscale scores by subsamples (site and trial participation status), generating scores by averaging item responses within each subscale.

## Ethical considerations

Ethical approval for this project was obtained from the FHI 360 Protection of Human Subjects Committee (protocol 1151358), the University of Cape Town Health Research Ethics Committee, and the Kenya Medical Research Institute Scientific and Ethics Review Unit. Written consent was obtained electronically from each participant before each survey. Data were de-identified to protect participant confidentiality.

## Results

### Sample characteristics

In general, the women who participated in our survey were young, with a mean age of 22 (Table 1). Women in South Africa were on average two years younger than women in Kenya and the majority (66%) lived with parents and other family members, whereas just over a third of Kenyan participants (35%) lived with parents. Just over one-quarter of South African women compared to almost one-half of Kenyans had at least one child. In both sites, former or current trial participants tended to be younger than women who had not participated in trials before. Most participants in both samples were sexually active with a regular partner, although they were not necessarily living with the partner. Women were eligible to participate in this study if they were currently or had been on a daily pill regimen, including daily contraceptive pills, daily vitamins or medications for another conditions, as well as PrEP. We included a wide range of products with the goal of validating our "oral daily pills" adherence items while also recognizing that some women who choose to join future prevention trials and would therefore respond to our scale items will never have used daily PrEP. Nevertheless, almost 80% had participated in trials previously and more than 80% of participants had used PrEP at some point in their past–either within a trial or as a prevention option outside of a trial.

### Screening tool

The root mean square error of approximation (RMSEA) for CFA model was 0.09, suggesting the 6-factor model was supported by the data using a 0.10 cut-off score. However, not all the goodness of fit statistics showed an acceptable fit (Table 2). Particularly, the Comparative Fit Index (CFI)–which should be at 0.90 or higher—was at inadequate levels (0.80 for the model on the full sample and 0.78 for the model on the FTP subset). To explore this further, we fit a separate CFA model to each one of the first five factors, studied the modification indices of the overall CFA fit, and re-did the analysis treating the item responses as continuous.

The factors Distrust of Research, Trial Incompatibility and Partner Disclosure were the best-fitting factors. When using the full sample, the goodness-of-fit statistics for the models on these factors were all at commonly-accepted levels (RMSEA upper bound below 0.10; CFI above 0.90). The factor Commitment to Research did not fit the data as well as other factors but was still at an acceptable level (RMSEA upper bound is high at 0.11; CFI above 0.90). For

**Table 1. Key socio-demographic information for survey sample.**

| | South Africa | | | Kenya | | | Total | | |
|---|---|---|---|---|---|---|---|---|---|
| | TNP | FTP | All | TNP | FTP | All | TNP | FTP | Sample |
| | (n = 8) | (n = 85) | (n = 93) | (n = 32) | (n = 68) | (n = 100) | (n = 40) | (n = 153) | (n = 193) |
| Age (years, mean) | 21.5 | 20.8 | 20.9 | 25.0 | 22.1 | 23.1 | 24.3 | 21.4 | 22.0 |
| Education (%) | | | | | | | | | |
| Primary or less | 0 | 7.1 | 6.5 | 18.2 | 18.0 | 18.1 | 13.3 | 11.2 | 11.5 |
| Some secondary | 0 | 30.6 | 28.0 | 22.7 | 12.0 | 15.3 | 16.7 | 23.7 | 22.4 |
| Secondary complete | 50.0 | 29.4 | 31.2 | 9.1 | 8.0 | 8.3 | 20.0 | 21.5 | 21.2 |
| College or higher | 50.0 | 33.0 | 34.4 | 50.0 | 62.0 | 58.3 | 50.0 | 43.7 | 44.9 |
| Relationship status (%) | | | | | | | | | |
| (Living as/) Married | 25.0 | 7.1 | 8.6 | 45.5 | 6.0 | 18.1 | 40.0 | 6.7 | 12.7 |
| Regular partner (not living together) | 50.0 | 68.2 | 66.7 | 50.0 | 58.0 | 55.6 | 50.0 | 64.4 | 61.8 |
| Sexually active, no regular partner(s) | 12.5 | 11.8 | 11.8 | 0 | 30.0 | 20.8 | 33.3 | 18.5 | 15.8 |
| Not sexually active | 12.5 | 12.9 | 12.9 | 4.6 | 6.0 | 5.6 | 6.7 | 10.4 | 9.7 |
| Currently living with (%) | | | | | | | | | |
| Parent/s | 75.0 | 64.7 | 65.6 | 36.4 | 34.0 | 34.7 | 46.7 | 53.3 | 52.1 |
| Other family | 0 | 22.4 | 20.4 | 9.1 | 12.0 | 11.1 | 6.7 | 18.5 | 16.4 |
| Partner | 12.5 | 2.4 | 3.2 | 40.9 | 12.0 | 20.8 | 33.3 | 5.9 | 10.9 |
| Children | 0 | 4.7 | 4.3 | 36.4 | 22.0 | 26.4 | 26.7 | 11.1 | 13.9 |
| Other | 12.5 | 7.1 | 7.5 | 13.6 | 36.0 | 29.2 | 13.3 | 17.8 | 17.0 |
| Have children (%) | 50.0 | 25.9 | 28.0 | 68.2 | 38.0 | 47.2 | 63.3 | 30.4 | 36.4 |
| Type of pill used+ (%) | | | | | | | | | |
| Oral contraceptive | 62.5 | 16.5 | 20.4 | 40.6 | 0 | 13.0 | 45.0 | 9.2 | 16.6 |
| Vitamin | 12.5 | 1.2 | 2.2 | 6.3) | 0 | 2.0 | 7.5 | 0.7 | 2.1 |
| Chronic health medication | 0 | 3.5 | 3.2 | 12.5 | 0 | 4.0 | 10.0 | 2.0 | 3.6 |
| PrEP | 25.0 | 78.8 | 74.2 | 40.6 | 100 | 81.0 | 37.5 | 88.2 | 77.7 |
| Used oral pills to prevent HIV (%) | 25.0 | 87.1 | 81.7 | 46.9 | 100 | 83.0 | 42.5 | 92.8 | 82.4 |
| Past trial participation (%) | | | 91.4 | | | 68.0 | | | 79.3 |
| Circumstances for PrEP use+ (%) | | | | | | | | | |
| Prescribed PEP | 0 | 2.4 | 2.2 | 0 | 0 | 0* | 0 | 1.3 | 1.0 |
| PrEP trial (past) | 37.5 | 42.4 | 41.9 | 0 | 0 | 0* | 7.5 | 23.5 | 20.2 |
| PrEP trial (current) | 25.0 | 48.2 | 46.2 | 0 | 100 | 68.0* | 5.0 | 71.2 | 57.5 |
| Other | 37.5 | 7.1 | 9.7 | 100 | 0 | 32.0* | 87.5 | 3.9 | 21.2 |

+More than one response possible

the Personal and Health Benefits factor, the CFI was low and the RMSEA was high, suggesting the model did not fit the data in this case.

## Monitoring tool

Items performed well overall, with only one item being dropped due to a ceiling effect. The remaining 44 items were included in EFAs. The scree plot showed six factors with eigenvalues of 2 or more before a relatively large drop to the seventh factor. In addition, the eigenvalue analysis indicated possible 3-, 4-, 5-, or 6-factor solutions.

Multiple possible solutions were assessed. Below we present the final set of factors and items (Table 3). We chose to use three factors from the 6-factor solution, as well as two factors from the 5-factor solution, because they aligned well with our previous vaginal ring monitoring tool and had acceptable psychometric properties.

**Table 2. Screening tool CFA model fit statistics.**

| Sample | Factor | N | N Items | Chi Sq. | RMSEA [95% CI] | CFI |
|---|---|---|---|---|---|---|
| All | All 6 factors | 179 | 40 | 1746.9*** | 0.09 [0.08, 0.09] | 0.80 |
| All | 1. Commitment to Research | 186 | 8 | 39.2** | 0.07 [0.04, 0.11] | 0.92 |
| All | 2. Personal & Health Benefits | 189 | 5 | 45.9*** | 0.21 [0.16, 0.27] | 0.87 |
| All | 3. Distrust of Research | 192 | 11 | 44.5 | 0.01 [0, 0.05] | 1.00 |
| All | 4. Trial Incompatibility | 186 | 8 | 20.4 | 0.01 [0, 0.06] | 1.00 |
| All | 5. Partner Disclosure | 191 | 5 | 3.33 | 0 [0, 0.08] | 1.00 |
| All | 6. Visit Adherence | | 3 | NA | NA | NA |
| FTP | All 6 factors | 140 | 38 | 1699.7*** | 0.10 [0.09, 0.11] | 0.78 |
| FTP | 1. Commitment to Research | 146 | 8 | 37.5*** | 0.08 [0.04, 0.12] | 0.91 |
| FTP | 2. Personal & Health Benefits | 150 | 7 | 61.4*** | 0.15 [0.11, 0.19] | 0.88 |
| FTP | 3. Distrust of Research | 152 | 10 | 36.8 | 0.02 [0.00, 0.06] | 1.00 |
| FTP | 4. Trial Incompatibility | 148 | 8 | 41.2*** | 0.08 [0.05, 0.12] | 0.80 |
| FTP | 5. Partner Disclosure | 151 | 5 | 1.49 | 0.00 [0.00, 0.04] | 1.00 |
| FTP | 6. Visit Adherence | | 3 | NA | NA | NA |

*p<0.10

**p<0.05

***p<0.01

Note: Results for factor Visit Adherence (Factor 6) not included as it only has 3 items.

Note: Results for the TNP sample not included due to small sample size (n = 40).

## Construct validity

Tables 4 and 5 below provide comparisons between the original screening and monitoring tools and the final proposed subscales from the current sample. We would expect for the reliability statistic on a scale to vary across samples. In general, these statistics are similar across the original and our current samples for both the screening tool and the monitoring tool, except in the case of the screening tool's Trial Incompatibility subscale and, to a lesser extent, the Partner Disclosure subscale.

In general, the correlations between screening and monitoring subscales in our new sample were similar in magnitude and direction as in our original sample (Table 6). Several exceptions are highlighted in Table 6.

Descriptive statistics for subscale scores are represented for each subscale in the selected solution in Table 7, disaggregated by site and trial participation status.

As reflected in Table 7, site level differences in mean scale scores were generally larger than any differences by clinical trial experience. South African participants scored significantly higher on most screening tool subscales than did Kenyan participants. These differences may reflect broader cultural factors, or may be due to differences in the socio-demographic variations in our study sample. As might be expected, former trial participants scored significantly higher on items related to Pill Benefits than women who had not participated in trials previously. There were no other differences in scale scores by trial experience.

## Discussion

The validity of psychosocial scales or measurement tools is strengthened to the degree that they perform well in multiple settings and over time [32]. To that end, this research study provided additional evidence that our previously developed screening and monitoring tools performed adequately or better in two new populations of potential and former clinical trial

**Table 3. Final (selected) solution derived from the 5- and 6-factor solutions.**

| Item | Loading |
|---|---|
| **Factor 1—Study Pill Challenges (Cronbach's α = 0.80)** | |
| 1. I am not always sure that I take the study pill correctly. | 0.67 |
| 2. Sometimes I feel that research staff think I don't understand how to use the study pill. | 0.62 |
| 3. I think there are times when the research staff is disappointed in me. | 0.60 |
| 4. Using the pill every day is too difficult. | 0.54 |
| 5. I sometimes have trouble taking the study pill because it is too large to swallow. | 0.52 |
| 6. Sometimes, I take the study pill earlier or later than normally. | 0.51 |
| 7. I sometimes forget to take the study pill for several days at a time. | 0.49 |
| 8. Sometimes, I only take the study pill a few days before coming for a clinic visit. | 0.45 |
| 9. I believe that there is a chance that the study pill might cause me harm. | 0.44 |
| 10. I sometimes forget to attend my research visits. | 0.44 |
| 11. Getting to the clinic appointments is a challenge. | 0.42 |
| 12. I have doubts that using the study pill will help me. | 0.41 |
| **Factor 2—Concerns about Side Effects (Cronbach's α = 0.78)** | |
| 1. Side effects make me want to stop using the study pill. | 0.81 |
| 2. I think about stopping use of the study pill when side effects begin to interfere with daily activities. | 0.72 |
| 3. The pill side effects interfere with my sex life. | 0.55 |
| 4. Study pill side effects interfere with my everyday life. | 0.53 |
| 5. Sometimes, if I feel worse when I take the study pill, I stop using it for a while. | 0.49 |
| 6. The study pill interferes with my sleep at night. | 0.44 |
| 7. I do not take the study pill when I do not feel well. | 0.43 |
| **Factor 3[*] - Positive Pill Adherence (Cronbach's α = 0.71)** | |
| 1. I am very confident that I can participate in research visits, even if I experience difficulties. | 0.70 |
| 2. I am able to deal with any problems in following my study requirements. | 0.60 |
| 3. I am very confident that I can attend clinic visits even if the visits interfere with my daily activities. | 0.59 |
| 4. Using the study pill to prevent HIV/pregnancy/illness is better than using condoms. | 0.53 |
| 5. The study pill makes my sex life better. | 0.51 |
| 6. There are more benefits of using the study pill than difficulties. | 0.51 |
| 7. Using the study pill makes me feel good about myself. | 0.50 |
| 8. I am sure I can discuss honestly with the clinic staff any problems related to the study pill. | 0.48 |
| 9. I am completely sure that I can continue to use the study pill on days when I do not feel well. | 0.43 |
| **Factor 4—Social Difficulties with Clinical Trial Participation (Cronbach's α = 0.73)** | |
| 1. It is difficult for me to explain the research to my family. | 0.82 |
| 2. It is difficult for me to explain the research to my friends. | 0.77 |
| 3. I do not want to be seen with the study pill container. | 0.63 |
| **Factor 5[*] - Pill Benefits (Cronbach's α = 0.68)** | |
| 1. I believe that the study pill will reduce my chance of getting HIV/pregnant/ill. | 0.74 |
| 2. I believe that my risk for HIV/pregnancy/illness is less when I am using the study pill. | 0.73 |
| 3. I believe that I might get HIV/pregnant/ill if I don't use the pill as instructed. | 0.47 |
| 4. I believe that my risk for HIV/pregnancy/illness is the same whether I used the study pill or not. [*Reverse scored*] | -0.64 |

[*]From 5-factor solution

participants in South Africa and Kenya. As oral PrEP clinical trials and public sector rollout of PrEP regimens continue in new settings and with new populations, having reliable tools to screen for potential adherence challenges and track adherence overtime will be vital to ensure that PrEP clinicians and providers can accurately follow product efficacy in these new contexts.

**Table 4. Comparison of screening tool subscales between sample of original creation and current sample.**

| Subscale | Number of Items | | Reliability (Cronbach's α) | | Item Example |
|---|---|---|---|---|---|
| | Original | Current | Original | Current | |
| Commitment to Research | 8 | 8 | 0.61 | 0.59 | "The idea of participating in research is appealing to me." |
| Personal & Health Benefits | 5 | 5 | 0.59 | 0.56 | "I want to participate in HIV prevention research because I want free health care." |
| Distrust of Research | 11 | 11 | 0.78 | 0.75 | "I do not trust research in general." |
| Trial Incompatibility | 8 | 8 | 0.69 | 0.51 | "People who participate in HIV prevention research may be rejected by others." |
| Partner Disclosure | 5 | 5 | 0.80 | 0.68 | "My partner knows that I am participating in the research." |
| Visit Adherence | 3 | 3 | 0.67 | 0.62 | "I have never been late for an appointment." |

These tools may add to their ability to do so; this study has further validated the screening tool and expanded the monitoring tool to a wider product scope by adding an oral PrEP-specific version. An important limitation to our current study is our inability to assess the predictive validity of our tools based on an objective biomarker of adherence in a prospective study design. We will evaluate the predictive validity of these tools as an exploratory objective in an upcoming PrEP trial in adolescents and young women in three African countries, launching in late 2021.

## Screening tool

When participants do not use products as instructed within trials, they reduce the potential of the trial to determine safety and efficacy. Although low adherence could indicate a lack of acceptability, the relationship between acceptability and adherence in clinical trial settings is further complicated by the trial context itself. For example, motivations to participate in order to receive reimbursements or access to health care, or the potential to be assigned to a placebo product or the need to adhere to additional trial-related procedures may impact trial participants' behaviors in ways that may not be assessed [33, 34]. Our original screening tool included six domains that could indicate a potential trial participants' propensity to adhere (9). A theoretical strength of the tool is that the items themselves are product-agnostic. When we evaluated the tool among women who may have participated in oral PrEP studies or had used daily pills for other purposes, the tool performed in a similar way to the original sample. While the "Visit Adherence" subscale was not further evaluated due to its brevity, three of five subscales

**Table 5. Comparison of monitoring tool subscales between sample of original creation and current sample.**

| Subscale | | Number of Items | | Reliability (Cronbach's α) | | Item Example | |
|---|---|---|---|---|---|---|---|
| Original | Current | Original | Current | Original | Current | Original | Current |
| Vaginal Ring Doubts | Study Pill Challenges | 8 | 12 | 0.78 | 0.80 | "It is hard to believe that using the vaginal ring will help me." | "I believe that there is a chance that the study pill might cause me harm." |
| Concerns about Side Effects | Concerns about Side Effects | 7 | 5 | 0.61 | 0.78 | "Side effects make it difficult for me to keep using the vaginal ring." | "Side effects make me want to stop using the study pill." |
| Vaginal Ring Removal | Positive Pill Adherence | 9 | 8 | 0.82 | 0.71 | "Sometimes, if I felt worse when I had the vaginal ring in my body, I stopped using it." | "I am completely sure that I can continue to use the study pill on days when I do not feel well." |
| N/A | Social Difficulties with Clinical Trial Participation | N/A | 3 | N/A | 0.73 | N/A | "It is difficult for me to explain the research to my family." |
| Vaginal Ring Benefits | Pill Benefits | 4 | 9 | 0.75 | 0.68 | "I believe that the vaginal ring will reduce my chance of getting HIV." | "I believe that the study pill will reduce my chance of getting HIV." |

**Table 6. Comparisons of current Pearson product moment correlations between screening tool and monitoring tool subscales with correlations from original sample.**

| Monitoring Tool Subscales | Screening Tool Subscales | | | | | |
| --- | --- | --- | --- | --- | --- | --- |
| | Commitment to Research | Personal & Health Benefits | Distrust of Research | Trial Incompatibility | Partner Disclosure | Visit Adherence |
| **Monitoring Tool Subscales** | | | | | | |
| Original Sample | | | | | | |
| Vaginal Ring Doubts | -0.04 | 0.05 | 0.42 | 0.67 | -0.16 | -0.35 |
| Concerns about Side Effects | -0.16 | 0.02 | 0.53 | 0.09* | -0.05 | -0.17 |
| Vaginal Ring Removal[a] | -0.04* | 0.03* | 0.21 | 0.09 | -0.12 | -0.25 |
| Vaginal Ring Benefits | 0.04 | 0.3 | -0.36* | 0.13 | 0.09 | 0.37 |
| Current Sample | | | | | | |
| Study Pill Challenges | 0.08 | 0.12 | 0.63 | 0.51 | -0.03 | -0.19 |
| Concerns about Side Effects | 0.01 | 0.09 | 0.38 | 0.34* | -0.05 | -0.16 |
| Positive Pill Adherence[a] | 0.41* | 0.29* | -0.25 | 0.04 | 0.24 | 0.24 |
| Social Difficulties with Clinical Trial Participation | -0.15 | -0.08 | 0.24 | 0.21 | -0.53 | -0.15 |
| Pill Benefits | 0.21 | 0.22 | 0.05* | 0.15 | 0.09 | 0.18 |

*Difference between Pearson product correlation in original sample and current sample is 0.20 or greater.

[a]Inverse relationship expected between these corresponding subscales because adherence is measured in the opposite direction.

showed good fit when compared to their previous performance–the "Distrust of Research," "Trial Incompatibility," and "Partner Disclosure" subscales. In contrast, two subscales–"Personal and Health Benefits" and "Commitment to Research"–had lower-than-desirable fit indices, suggesting that participants in this sample may have understood and/or responded to these items differently than in the original study. Further research could be beneficial to refine item content. Nevertheless, the correlations between all of the subscales were consistent with our

**Table 7. Mean scores for screening tool and monitoring tool subscales by site and trial experience.**

| | Site | | Trial Experience | | Full Sample |
| --- | --- | --- | --- | --- | --- |
| | South Africa | Kenya | FTP | NTP | |
| Screening Tool | | | | | |
| Commitment to Research | 5.0 | 4.9 | 5.0 | 5.1 | 5.0 |
| Personal & Health Benefits | 5.2*** | 4.8*** | 5.0 | 5.1 | 5.0 |
| Distrust of Research | 2.7*** | 2.1*** | 2.4 | 2.4 | 2.4 |
| Trial Incompatibility | 3.4*** | 2.9*** | 3.1 | 3.2 | 3.2 |
| Partner Disclosure | 5.0*** | 4.3*** | 4.6 | 4.5 | 4.6 |
| Visit Adherence | 4.9 | 5.0 | 4.9 | 5.0 | 4.9 |
| Monitoring Tool | | | | | |
| Study Pill Challenges | 3.1*** | 2.5*** | 2.8 | 2.7 | 2.8 |
| Concerns about Side Effects | 2.5 | 2.3 | 2.3 | 2.7 | 2.4 |
| Positive Pill Adherence | 4.9 | 5.0 | 5.0 | 4.6 | 5.0 |
| Social Difficulties with Clinical Trial Participation | 2.3*** | 3.4*** | 2.7 | 3.3 | 2.8 |
| Pill Benefits | 5.1* | 4.8* | 5.1*** | 3.8*** | 4.9 |

*p<0.10 between subgroups

**p<0.05

***p<0.01

original assumptions based on the literature [9], providing evidence for their construct validity. An additional limitation is that our tools (both the original vaginal ring and oral PrEP scales) were developed and validated among young adult African women from South Africa and Kenya. Additional studies may be needed to validate their use among adolescent women, men or in other country contexts.

A validated screening tool could be a useful way to flag potential challenges related to product adherence in clinical trial research that stem from trial distrust, partner disclosure or misalignments in personal motivations or life contexts that undermine product use. With some adaptation of items to reflect a service delivery rather than clinical trial setting, some domains (e.g., distrust, partner disclosure, visit adherence) might be useful to assess a potential client's challenges to PrEP adherence–or to determine whether clients can be provided with a greater number of PrEP pill cycles between routine visits. Furthermore, with the approval of long-acting products, which are less dependent on user adherence, some screening scales could aid in the shared decision-making and choice of PrEP methods.

## Monitoring tool

This study also provided further support for the structure and content of an oral pill monitoring tool. The monitoring tool items, which were revised to assess perceptions about and experiences using daily oral PrEP, factored into five subscales that showed moderate to good reliability. Four of the five subscales had a similar structure overall to the vaginal ring monitoring tool items from which they were adapted. "Study Pill Challenges" captured similar difficulties in adherence as "Vaginal Ring Doubts." "Pill Benefits" likewise captured benefits to product use similarly to "Vaginal Ring Benefits." "Concerns about Side Effects" is nearly product-agnostic and captures a range of concerns applicable to any product. The subscales "Vaginal Ring Removal" and "Positive Pill Adherence" demonstrated the greatest change between products; however, this is expected given the difference between the "active" adherence required of oral PrEP in comparison to the more "passive" adherence of vaginal rings [29, 35, 36]. An additional subscale related to "Social Difficulties of Trial Participation" emerged from the EFA. Given trial participants' and PrEP users' reported challenges to adherence due to perceived stigma and issues with disclosure, we believe these items would be useful to monitor over the course of trial participation. In non-trial settings, clients' assessments of product-related benefits, challenges or adherence-related behaviors, self-administered at the clinic or via an app, could monitor changes in adherence determinants or guide delivery of more targeted counseling or communications messages aimed at supporting adherence prospectively.

## Conclusions

Accurately measuring product adherence in HIV-prevention clinical trials and non-trial settings is vitally important in understanding and supporting product efficacy and safety. The effectiveness of a HIV PrEP intervention depends on adequate adherence. Even as new biomarker measures to monitor adherence are developed, measures that explore adherence motivations and challenges in addition to actual product adherence will help researchers develop and bring new products to scale and enhance clinicians' capacity to support product use. Behavioral assessments such as the screening and monitoring tools discussed here are inexpensive, non-invasive, and allow for immediate feedback and the construction of a matrix of different factors of adherence and are likely to continue to be used widely in conjunction with biomarker adherence measures. In this study, we provided further evidence of validation for these measures, demonstrating the screening tool's utility in additional populations and adapting the monitoring tool's utility for different HIV-prevention products. To date, we have not

had the opportunity to assess the ability of the screening and monitoring tools to predict and monitor adherence prospectively. However, we have plans to do this in an upcoming trial. We also invite other clinical researchers and program implementers to assist in prospectively validating some or all of the domains in the screening and monitoring tools by evaluating their ability to predict adherence patterns as measured by objective biomarkers or other adherence measures in future clinical trials or PrEP-related programs.

## Acknowledgments

We would like to acknowledge the willing contributions of the women who participated in this study without whom this study would not have been possible. This project was made possible by the generous assistance from the American people through the U.S. President's Emergency Plan for AIDS Relief (PEPFAR) and the U.S. Agency for International Development (USAID). The contents are the responsibility of the authors and do not necessarily reflect the views of PEPFAR, USAID or the United States Government.

## Author Contributions

**Conceptualization:** Elizabeth E. Tolley, Gustavo F. Doncel.

**Data curation:** Katherine Gill, Nelly Mugo, Laura Myers, Ednar Casmir, Menna Duyver, Kenneth Ngure.

**Formal analysis:** Seth Zissette, Elizabeth E. Tolley, Andres Martinez.

**Funding acquisition:** Elizabeth E. Tolley, Homaira Hanif, Gustavo F. Doncel.

**Investigation:** Katherine Gill, Laura Myers, Ednar Casmir, Menna Duyver, Kenneth Ngure.

**Methodology:** Elizabeth E. Tolley, Andres Martinez.

**Project administration:** Seth Zissette, Laura Myers, Menna Duyver.

**Software:** Seth Zissette.

**Supervision:** Homaira Hanif, Katherine Gill, Nelly Mugo, Kenneth Ngure.

**Validation:** Andres Martinez.

**Visualization:** Seth Zissette, Andres Martinez.

**Writing – original draft:** Seth Zissette, Elizabeth E. Tolley.

**Writing – review & editing:** Andres Martinez, Homaira Hanif, Katherine Gill, Nelly Mugo, Kenneth Ngure, Gustavo F. Doncel.

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
