## [Decision Letter · Decision Letter 0]

12 Apr 2021

PONE-D-21-03228

Adaptation and validation of simple tools to screen and monitor for oral PrEP adherence

PLOS ONE

Dear Dr. Tolley,

Thank you for submitting your manuscript to PLOS ONE. After careful consideration, we feel that it has merit but does not fully meet PLOS ONE’s publication criteria as it currently stands. Therefore, we invite you to submit a revised version of the manuscript that addresses the points raised during the review process.

We look forward to receiving your revised manuscript.

Kind regards,

José das Neves

Academic Editor

PLOS ONE

Journal Requirements:

2. Please remove the "DRAFT" that you currently have showing as a watermark in the background of the manuscript pages.

Reviewers' comments:

Reviewer's Responses to Questions

**Comments to the Author**

1. Is the manuscript technically sound, and do the data support the conclusions?

Reviewer #1: Yes

Reviewer #2: Yes

2. Has the statistical analysis been performed appropriately and rigorously? 

Reviewer #1: Yes

Reviewer #2: Yes

3. Have the authors made all data underlying the findings in their manuscript fully available?

Reviewer #1: Yes

Reviewer #2: Yes

4. Is the manuscript presented in an intelligible fashion and written in standard English?

Reviewer #1: Yes

Reviewer #2: Yes

5. Review Comments to the Author

Reviewer #1: The paper touches on an important topic regarding product adherence in clinical trials. The creation of a simple tool that can be used in low resource settings in commendable. The paper however would benefit from a clearer representation of the process undertaken, which could be remedied by a simple schematic to orientate the reader.

Reviewer #2: This is an interesting analysis from Tolley and colleagues. I am not an expert on the validation of psychometric scales so I will focus on contextualization and framing of the findings. In spite of the significant challenges that survey adherence scales have experienced in the field of HIV PrEP--I agree they still play an important role. The paper is well written although I think the methods could use some clarification and some additional attention to contextualizing the findings.

Major point 1: Methods: Please provide additional detail in the methods section about the site, design, and context of the original vaginal ring monitoring tool.

2. Please also clarify the use of the tool for other product types (re: recruitment of product naive participants). What about many women who have not taken a daily pill prior to PrEP? Would this tool still be administered.

3. Intro: You provide a reasonable discussion of the pros and cons of objective vs. self-reported metrics. Is there data that demonstrates that the ring survey correlates with objective adherence? If so I would include it in the introduction or discussion as justification for adapting the scale. Furthermore, ideally this scale would be studied in conjunction with objective adherence metrics as further validation. This will be the major hurdle, given what the field has since learned, in uptake of such a scale.

4. Intro: You mention event-drive PrEP but your scale is not set up to address this delivery strategy nor is it currently indicated in this context so I would remove it. White coat adherence is an issue for short-term adherence markers, cumulative metrics are not impacted.

5. Page 24 line 334. This tool has been presented as a study product monitoring tool, it is not clear to me how it could be used as part of shared decision making between long-acting and other PrEP methods.

6. Page 25 Line 357 and Abstract Conclusions, as no objective adherence metric was included, I don't think you should focus on accuracy during the discussion. I found this confusing when i first read the abstract. I would mention correlation with objective adherence metrics as a next step/limitation

7. Abstract: Need to find space in methods that you adapted the ring tool to measure oral PrEP adherence

Minor

1: Discussion: Page 22 line 306. I am not sure that the relationship between adherence and acceptability is "confounded" by the trial design, although the relationship is more complex and has additional inputs. I would try to reformulate this statement to be more clear

6. PLOS authors have the option to publish the peer review history of their article (what does this mean?). If published, this will include your full peer review and any attached files.

Reviewer #1: No

Reviewer #2: No

---

## [Author Response · Author response to Decision Letter 0]

19 Apr 2021

2. Please remove the "DRAFT" that you currently have showing as a watermark in the background of the manuscript pages. 

Response: This has been done. 

Response: We have added the link and will activate it once the paper has been published.

Reviewer's Responses to Questions

5. Review Comments to the Author

Reviewer #1: The paper touches on an important topic regarding product adherence in clinical trials. The creation of a simple tool that can be used in low resource settings in commendable. The paper however would benefit from a clearer representation of the process undertaken, which could be remedied by a simple schematic to orientate the reader.

Response: We clarified the relationship between the “original” study, which was described in the introduction, but had not been directly identified. Information about the original study sample and process can be found in reference 9, which was published in PLoS One in 2018. We believe that the request for a schematic was likely related to confusion about whether the “original” study was separate from, or part of the study described in the manuscript. We hope that our track-changed edits have resolved this lack of clarity.

Additional comments from reviewer 1: Please provide information for how participants recruited.

Response: We have now provided that information. 

What was meant by “uniformly”?

Response: We clarified that, although randomly ordered, the same screening items were administered to all participants (unlike the monitoring items, which were either focused on oral PrEP or other daily oral pills.)

Provide description of the Likert scale.

Response: We provided labels for all 6 points on the scale. 

Provide information about the “original study” and the “cognitive interviews”.

Response: We have added some additional information about the cognitive interviews in the methods. This was a preliminary step to ensure that the screening and monitoring scales were relevant, comprehensible and easy to understand by women in both sites. We do not present additional information on this step because we did not make any major alternations to the items based on this research step. 

Response: 

Reviewer #2: This is an interesting analysis from Tolley and colleagues. I am not an expert on the validation of psychometric scales so I will focus on contextualization and framing of the findings. In spite of the significant challenges that survey adherence scales have experienced in the field of HIV PrEP--I agree they still play an important role. The paper is well written although I think the methods could use some clarification and some additional attention to contextualizing the findings.

Major point 1: Methods: Please provide additional detail in the methods section about the site, design, and context of the original vaginal ring monitoring tool.

Response: We are concerned about adding too much additional information about the original study in this manuscript and hope that further clarification linking the last two paragraphs of the introduction section will be sufficient. In addition, we have provided the reference to the original study in this section so that readers can find information about the study design, sample and the initial and finalized items for the two tools.

2. Please also clarify the use of the tool for other product types (re: recruitment of product naive participants). What about many women who have not taken a daily pill prior to PrEP? Would this tool still be administered.

Response: The monitoring tool is intended to work with any women who begin using a daily PrEP regimen within a trial setting or in the context of routine care. However, if she were using a different type of product (e.g., vaginal ring or injectable), the monitoring items – because they are product-specific – would need to change. Our original study validated a set of monitoring items that were focused on vaginal ring use. However, we have not developed items related to injections, for example. 

3. Intro: You provide a reasonable discussion of the pros and cons of objective vs. self-reported metrics. Is there data that demonstrates that the ring survey correlates with objective adherence? If so I would include it in the introduction or discussion as justification for adapting the scale. Furthermore, ideally this scale would be studied in conjunction with objective adherence metrics as further validation. This will be the major hurdle, given what the field has since learned, in uptake of such a scale.

Response: We agree with the reviewer. In fact, the original impetus of the study was to adapt the monitoring tool so that it could be used – and prospectively validated in trials of new PrEP regimens. We are now able to reference in the Conclusion section an upcoming phase 2 trial of Descovy versus Truvada in which we have added an exploratory objective aimed at prospectively validating our tools. We also hope that other clinical researchers will also consider prospectively evaluating the relationship of these tools with objective measures of adherence. 

4. Intro: You mention event-drive PrEP but your scale is not set up to address this delivery strategy nor is it currently indicated in this context so I would remove it. White coat adherence is an issue for short-term adherence markers, cumulative metrics are not impacted.

Response: Thank you for this comment. We have removed the reference to on-demand PrEP, but also linked the existing references to the hair assay which is currently one of the most promising assays for cumulative adherence. 

5. Page 24 line 334. This tool has been presented as a study product monitoring tool, it is not clear to me how it could be used as part of shared decision making between long-acting and other PrEP methods.

Response: The screening scales are product-agnostic. However, as the reviewer points out, they are currently oriented towards clinical trial rather than service delivery contexts. Although additional adaptation would be required, some scales – including distrust of product and trial contexts and partner disclosure – are factors influencing adherence to PrEP in routine service contexts. We have modified the text to reflect the need for some adaptation. 

6. Page 25 Line 357 and Abstract Conclusions, as no objective adherence metric was included, I don't think you should focus on accuracy during the discussion. I found this confusing when i first read the abstract. I would mention correlation with objective adherence metrics as a next step/limitation

Response: We’d like to thank the reviewer for highlighting this point. We have added this as a limitation in the discussion and noted that we have plans to assess the predictive validity of the tool in assessing adherence via objective biomarker. We have also invited other clinical scientists to assist in the tools’ prospective assessment in the conclusion section of the manuscript. 

7. Abstract: Need to find space in methods that you adapted the ring tool to measure oral PrEP adherence

Response: We feel we have included this information already in the abstract, which states in the Introduction: 

In a previous study, we developed tools to screen for general adherence and specifically monitor intravaginal ring adherence within the context of HIV prevention clinical trials. This study aimed to further validate the screening tool and to adapt and provide initial psychometric validation for an oral pill monitoring tool.

Discission P22 line 306: I am not sure that the relationship between adherence and acceptability is "confounded" by the trial design, although the relationship is more complex and has additional inputs. I would try to reformulate this statement to be more clear.

Response: Thank you for this comment. We have reworded to clarify that “the relationship … is further complicated by the trial context itself.” 

---

## [Decision Letter · Decision Letter 1]

4 May 2021

Adaptation and validation of simple tools to screen and monitor for oral PrEP adherence

PONE-D-21-03228R1

Dear Dr. Tolley,

We’re pleased to inform you that your manuscript has been judged scientifically suitable for publication and will be formally accepted for publication once it meets all outstanding technical requirements.

Kind regards,

José das Neves

Academic Editor

PLOS ONE

Additional Editor Comments (optional):

Reviewers' comments:

Reviewer's Responses to Questions

**Comments to the Author**

1. If the authors have adequately addressed your comments raised in a previous round of review and you feel that this manuscript is now acceptable for publication, you may indicate that here to bypass the “Comments to the Author” section, enter your conflict of interest statement in the “Confidential to Editor” section, and submit your "Accept" recommendation.

Reviewer #2: All comments have been addressed

2. Is the manuscript technically sound, and do the data support the conclusions?

Reviewer #2: Yes

3. Has the statistical analysis been performed appropriately and rigorously? 

Reviewer #2: Yes

4. Have the authors made all data underlying the findings in their manuscript fully available?

Reviewer #2: No

5. Is the manuscript presented in an intelligible fashion and written in standard English?

Reviewer #2: Yes

6. Review Comments to the Author

Reviewer #2: The authors have addressed my concerns adequately. I agree that describing the initial trial would be laborious but some clarification would help the reader. For point 7 from original reviewer 2, I think it would be helpful to state this point not just in the abstract, but also in text, as these should be considered independent products.

7. PLOS authors have the option to publish the peer review history of their article (what does this mean?). If published, this will include your full peer review and any attached files.

Reviewer #2: No

---

## [Editor Report · Acceptance letter]

19 May 2021

PONE-D-21-03228R1 

Adaptation and validation of simple tools to screen and monitor for oral PrEP adherence

Dear Dr. Tolley:

I'm pleased to inform you that your manuscript has been deemed suitable for publication in PLOS ONE. Congratulations! Your manuscript is now with our production department. 

Kind regards, 

on behalf of

Dr. José das Neves 

Academic Editor

PLOS ONE